# INTER-BMV:
# INTERPOLATION WITH BLOCK MOTION VECTORS FOR FAST SEMANTIC SEGMENTATION ON VIDEO

## ABSTRACT

Models optimized for accuracy on single images are often prohibitively slow to run on each frame in a video. Recent work exploits the use of optical flow to warp image features forward from select keyframes, as a means to conserve computation on video. This approach, however, achieves only limited speedup, even when optimized, due to the accuracy degradation introduced by repeated forward warping, and the inference cost of optical flow estimation. To address these problems, we propose a new scheme that propagates features using the block motion vectors (BMV) present in compressed video (e.g. H.264 codecs), instead of optical flow, and bi-directionally warps and fuses features from enclosing keyframes to capture scene context on each video frame. Our technique, interpolation-BMV, enables us to accurately estimate the features of intermediate frames, while keeping inference costs low. We evaluate our system on the CamVid and Cityscapes datasets, comparing to both a strong single-frame baseline and related work. We find that we are able to substantially accelerate segmentation on video, achieving near real-time frame rates (20+ frames per second) on large images (e.g. $960 \times 720$ pixels), while maintaining competitive accuracy. This represents an improvement of almost $6\times$ over the single-frame baseline and $2.5\times$ over the fastest prior work.

## 1 INTRODUCTION

Semantic segmentation, the task of assigning each pixel in an image to a semantic object class, is a problem of long-standing interest in computer vision. Like models for other image recognition tasks (e.g. classification, detection, instance segmentation), semantic segmentation networks have grown drastically in both layer depth and parameter count in recent years, in the race to segment more complex images, from larger, more realistic datasets, at higher accuracy. As a result, state-of-the-art segmentation networks today require between 0.5 to 3.0 seconds to segment a *single*, high-resolution image (e.g. $2048 \times 1024$ pixels) at competitive accuracy (Zhu et al. (2017); Gadde et al. (2017)).

Meanwhile, a new target data format for segmentation has emerged: video. The motivating use cases include both batch applications, where video is segmented in bulk to generate training data for other models (e.g. autonomous control systems), and streaming settings, where high-throughput video segmentation enables interactive analysis of live footage (e.g. at surveillance sites). Video in these contexts consists of long image sequences, shot at high frame rates (e.g. 30 fps) in complex environments (e.g. urban cityscapes) on modern, high-definition cameras. Segmenting individual frames at high accuracy still calls for the use of competitive image models, but their inference cost precludes their naïve deployment on every frame in a raw multi-hour video stream.

A defining characteristic of realistic video is its high level of temporal continuity. Consecutive frames demonstrate significant spatial similarity, which suggests the potential to reuse computation across frames. Building on prior work, we exploit two observations: 1) higher-level features evolve more slowly than raw pixel content in video, and 2) feature computation tends to be much more expensive than task-specific computation across a range of vision tasks (e.g. detection, segmentation) (Shelhamer et al. (2016); Zhu et al. (2017)). Accordingly, we divide our semantic segmentation model into a deep *feature network* and a cheap, shallow *task network* (Zhu et al. (2017)). We compute features only on designated keyframes, and propagate them to intermediate frames, by warping the feature maps with frame-to-frame motion estimates. The task network is executed on all frames.

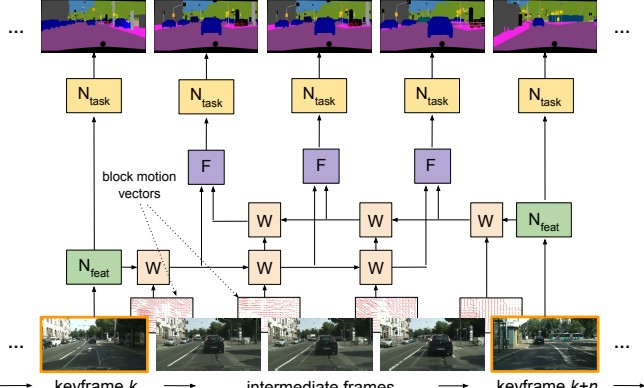

Figure 1: Inter-BMV warps and fuses the features of enclosing keyframes to generate accurate feature estimates for intermediate frames, using block motion vectors present in compressed video.

Given that feature warping and task computation is much cheaper than feature extraction, a key parameter we aim to optimize is the interval between designated keyframes.

Here we make two key contributions. First, noting the high level of data redundancy in video, we successfully utilize an artifact of compressed video, block motion vectors (BMV), to cheaply propagate features from frame to frame. Unlike other motion estimation techniques, which require specialized convolutional networks, block motion vectors are freely available in modern video formats, making for a simple, fast design. Second, we propose a novel feature estimation technique that enables the features for a large fraction of video frames to be inferred accurately and efficiently (see Fig. 1). In particular, when computing the segmentation for a keyframe, we also precompute the features for the *next* designated keyframe. Features for all subsequent intermediate frames are then computed as a *fusion* of features warped forward from the last visited keyframe, and features warped backward from the incoming keyframe. This procedure implements an *interpolation* of the features of the two closest keyframes. We then combine the two ideas, using block motion vectors to perform the feature warping in feature interpolation. The result is a scheme we call **interpolation-BMV**.

We evaluate our framework on the CamVid and Cityscapes datasets. Our baseline consists of running a competitive segmentation network, DeepLab (Chen et al. (2017)), on every frame, a setup that achieves published accuracy (Dai et al. (2017)), and throughput of 3.6 frames per second (fps) on CamVid and 1.3 fps on Cityscapes. Our improvements come in two phases. First, our use of motion vectors for feature propagation allow us to cut inference time on intermediate frames by 53%, compared to approaches based on optical-flow, such as Zhu et al. (2017). Second, our bi-directional feature warping and fusion scheme achieves substantial accuracy improvements, especially at high keyframe intervals. Together, the two techniques allow us to operate at over twice the average inference speed as the fastest prior work, at any target level of accuracy. For example, if we are willing to tolerate no worse than 65 mIoU on our CamVid video stream, we are able to operate at a throughput of 20.1 fps, compared to the 8.0 fps achieved by the forward flow-based propagation from Zhu et al. (2017). Overall, even when operating in high accuracy regimes (e.g. within 3% mIoU of the baseline), we are able to accelerate segmentation on video by a factor of 2-6×.

## 2    RELATED WORK

### 2.1    IMAGE SEMANTIC SEGMENTATION

Semantic segmentation is a classical image recognition task in computer vision, originally studied in the context of statistical inference. The approach of choice was to propagate evidence about pixel class assignments through a probabilistic graphical model (Felzenszwalb & Huttenlocher (2004); Shotton et al. (2009)), a technique that scaled poorly to large images with numerous object classes (Krähenbühl & Koltun (2011)). In 2014, Long et al. (2015) proposed the use of fully convolutional neural networks (FCNs) to segment images, demonstrating significant accuracy gains on several key datasets. Subsequent work embraced the FCN architecture, proposing augmentations such as

dilated (atrous) convolutions (Yu & Koltun (2016)), post-processing CRFs (Chen et al. (2016)), and pyramid spatial pooling (Zhao et al. (2017)) to further improve accuracy on large, complex images.

## 2.2 EFFICIENT VIDEO SEMANTIC SEGMENTATION

The recent rise of applications such as autonomous driving, industrial robotics, and automated video surveillance, where agents must perceive and understand the visual world *as it evolves*, has triggered substantial interest in the problem of efficient video semantic segmentation. Shelhamer et al. (2016) and Zhu et al. (2017) proposed basic feature reuse and optical flow-based feature warping, respectively, to reduce the inference cost of running expensive image segmentation models on video. Recent work explores adaptive feature propagation, partial feature updating, and adaptive keyframe selection as techniques to further optimize the scheduling and execution of optical-flow based warping (Zhu et al. (2018); Li et al. (2018); Xu et al. (2018)). In general, these techniques fall short in two respects: (1) optical flow computation remains a computational bottleneck, especially as other network components become cheaper, and (2) forward feature propagation fails to account for other forms of temporal change, besides spatial displacement, such as new scene content (e.g. new objects), perspective changes (e.g. camera pans), and observer movement (e.g. in driving footage). As a result, full frame features must still be recomputed frequently to maintain accuracy, especially in video footage with complex dynamics, fundamentally limiting the attainable speedup.

## 2.3 MOTION AND COMPRESSED VIDEO

Wu et al. (2018) train a network directly on compressed video to improve both accuracy and performance on video action recognition. Zhang et al. (2016) replace the optical flow network in the classical two-stream architecture (Simonyan & Zisserman (2014)) with a "motion vector CNN", but encounter accuracy challenges, which they address with various transfer learning schemes. Unlike these works, our main focus is not efficient training, nor reducing the physical size of input data to strengthen the underlying signal for video-level tasks, such as action recognition. We instead focus on a class of dense prediction tasks, notably semantic segmentation, that involve high-dimensional output (e.g. a class prediction for every pixel in an image) generated on the original uncompressed frames of a video. This means that we must still process each frame in isolation. To the best of our knowledge, we are the first to propose the use of compressed video artifacts to warp deep neural representations, with the goal of drastically improved inference throughput on realistic video.

## 3 SYSTEM OVERVIEW

### 3.1 NETWORK ARCHITECTURE

We follow the common practice of adapting a competitive image classification model (e.g. ResNet-101) into a fully convolutional network trained on the semantic segmentation task (Long et al. (2015); Yu et al. (2017); Chen et al. (2017)). We identify two logical components in our final model: a *feature network*, which takes as input an image $i \in R^{1 \times 3 \times h \times w}$ and outputs a representation $f_i \in R^{1 \times A \times \frac{h}{16} \times \frac{w}{16}}$, and a *task network*, which given the representation, computes class predictions for each pixel in the image, $p_i \in R^{1 \times C \times h \times w}$. The task network $N_{task}$ is built by concatenating three blocks: (1) a feature projection block, which reduces the feature channel dimensionality to $\frac{A}{2}$, (2) a scoring block, which predicts scores for each of the $C$ segmentation classes, and (3) an upsampling block, which bilinearly upsamples the score maps to the resolution of the input image.

### 3.2 BLOCK MOTION VECTORS

MPEG-compressed video consists of two logical components: reference frames, called I-frames, and delta frames, called P-frames. Reference frames are still RGB frames from the video, usually represented as spatially-compressed JPEG images. Delta frames, which introduce temporal compression to video, consist of two subcomponents: block motion vectors and residuals.

Block motion vectors, the artifact of interest in our current work, define a correspondence between pixels in the current frame and pixels in the previous frame. They are generated using *block motion compensation*, a standard procedure in video compression algorithms (Richardson (2008)):

1. Divide the current frame into a non-overlapping grid of 16x16 pixel blocks.

2. For each block in the current frame, determine the "best matching" block in the previous frame. A common matching metric is to minimize mean squared error between the blocks.

3. For each block in the current frame, represent the pixel offset to the best matching block in the previous frame as an $(x, y)$ coordinate pair, or *motion vector*.

The resulting grid of $(x, y)$ offsets forms the *block motion vector map* for the current frame. For a $16M \times 16N$ frame, this map has dimensions $M \times N$. The residuals then consist of the pixel-level difference between the current frame, and the previous frame transformed by the motion vectors.

### 3.3 FEATURE PROPAGATION

Many cameras compress video by default as a means for efficient storage and transmission. The availability of a free form of motion estimation at inference time, the motion vector maps in MPEG-compressed video, suggests the following scheme for fast video segmentation (see **Algorithm 1**).

---

**Algorithm 1** Feature propagation with block motion vectors (**prop-BMV**)

---
1: **input**: video frames $\{I_i\}$, motion vectors mv, keyframe interval $n$
2: **for** frame $I_i$ **in** $\{I_i\}$ **do**
3:     **if** i **mod** n = 0 **then**                                   ▷ keyframe
4:         $f_i \leftarrow N_{feat}(I_i)$                         ▷ keyframe features
5:         $S_i \leftarrow N_{task}(f_i)$
6:     **else**                                    ▷ intermediate frame
7:         $f_i \leftarrow \text{WARP}(f_c, -\text{mv}[i]))$         ▷ warp cached features
8:         $S_i \leftarrow N_{task}(f_i)$
9:     **end if**
10:    $f_c \leftarrow f_i$                               ▷ cache features
11: **end for**
12: **output**: frame segmentations $\{S_i\}$

---

Choose a keyframe interval $n$. On keyframes (every $n^{\text{th}}$ frame), execute the feature network $N_{feat}$ to obtain a feature map. Cache these computed features, $f_c$, and then execute the task network $N_{task}$ to obtain the keyframe segmentation. On intermediate frames, extract the motion vectors $\text{mv}[i]$ corresponding to the current frame index. Warp the cached features $f_c$ one frame forward via bilinear interpolation with $-\text{mv}[i]$. (To warp forward, we apply the negation of the vector map.) Here we employ the differentiable, parameter-free spatial warping operator proposed by Jaderberg et al. (2015). Finally, execute $N_{task}$ on the warped features to obtain the current segmentation.

### 3.3.1 INFERENCE RUNTIME ANALYSIS

Feature propagation is effective because it relegates feature extraction, the most expensive network component, to select keyframes. Of the three remaining operations performed on intermediate frames – motion estimation, feature warping, and task execution – motion estimation with optical flow is the most expensive (see Fig. 2). By using block motion, we eliminate this remaining bottleneck, accelerating inference times on intermediate frames for a DeepLab segmentation network (Chen et al. (2017)) from 116 ms per frame ($F + W + N_{task}$) to 54 ms per frame ($W + N_{task}$). For keyframe interval $n$, this translates to a speedup of 53% on $\frac{n-1}{n}$ of the video frames.

Note that for a given keyframe interval $n$, as we reduce inference time on intermediate frames to zero, we approach a maximum attainable speedup factor of $n$ over a frame-by-frame baseline that runs the full model on every frame. Exceeding this bound, without compromising on accuracy, requires an entirely new approach to feature estimation, the subject of the next section.

Incidentally, we also benchmarked the time required to extract block motion vectors from raw video (i.e. H.264 compression time), and found that `ffmpeg` takes 2.78 seconds to compress 1,000 Cityscapes video frames, or 2.78 ms per frame. In contrast, optical flow computation on a frame pair takes 62 ms (Fig. 2). We include this comparison for completeness: since compression is a default behavior on modern cameras, block motion extraction is not a true component of inference time.

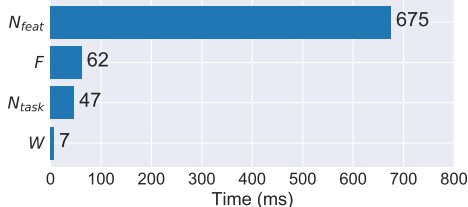

Figure 2: A sample runtime breakdown for a ResNet-101 DeepLab network. $F$ is the optical flow net from Dosovitskiy et al. (2015). $W$ is the warp operator. GPU: Tesla K80. Dataset: Cityscapes.

## 3.4 FEATURE INTERPOLATION

Given an input video stream, our goal is to compute the segmentation of every frame as efficiently as possible, while preserving accuracy. In a batch setting, we have access to the entire video, and desire the segmentations for all the frames, as input to another model (e.g. an autonomous control system). In a streaming setting, we have access to frames as they come in, but may be willing to tolerate a small delay of keyframe interval $n$ frames ($\frac{n}{30}$ seconds at 30 fps) before we output a segmentation, if that means we can match the throughput of the video stream and maintain high accuracy.

We make two observations. First, all intermediate frames in a video by definition lie between two designated keyframes, which represent bounds on the current scene. New objects that are missed in forward feature propagation schemes are more likely to be captured if both past and incoming keyframes are used. Second, feature fusion techniques are effective at preserving strong signals in any one input feature map, as seen in Feichtenhofer et al. (2016). This suggests the viability of estimating the features of intermediate frames as the fusion of the features of enclosing keyframes.

Expanding on this idea, we propose the following algorithm (see Fig. 1). On any given keyframe, precompute the features for the *next* keyframe. On intermediate frames, warp the previous keyframe's features, $N_{feat}(I_k)$, forward to the current frame $I_i$ using incremental forward motion estimates, $-\text{mv}[k:i]$. Warp the next keyframe's features, $N_{feat}(I_{k+n})$, *backward* to the current frame using incremental backward motion estimates, $\text{mv}[k+n:i]$. Fuse the two feature maps using either a weighted average or learned fusion operator, $F$. Then execute the task network $N_{task}$ on the fused features. This forms **Algorithm 2**. A formal statement is included in Appendix: Sec. 6.1.

To eliminate redundant computation, on keyframes, we *precompute* forward and backward warped feature maps $f^f, f^b$ corresponding to each subsequent intermediate frame, $\{I_{k+1}, ..., I_{k+n-1}\}$. For keyframe interval $n$, this amounts to $n-1$ forward and $n-1$ backward warped feature maps.

### 3.4.1 FEATURE FUSION

We consider several possible fusion operators: max fusion, average fusion, and convolutional fusion (Feichtenhofer et al. (2016)). We implement max and average fusion by aligning the input feature maps $f^f, f^b \in R^{1 \times C \times h \times w}$ along the channel dimension, and computing a max or average across each pixel in corresponding channels, a parameter-free operation. We implement conv fusion by *stacking* the input feature maps along the channel dimension $[f^f, f^b]_C = f^s \in R^{1 \times 2C \times h \times w}$, and applying a bank of learned, 1x1 conv filters to reduce the channel dimensionality by a factor of two.

Before applying the fusion operator, we weight the two input feature maps $f^f, f^b$ by scalars $\alpha$ and $1 - \alpha$, respectively, that correspond to *feature relevance*, a scheme that works very effectively in practice. For keyframe interval $n$, and a frame at offsets $p$ and $n-p$ from the previous and next keyframes, respectively, we set $\alpha = \frac{n-p}{n}$ and $1 - \alpha = \frac{p}{n}$, thereby penalizing the input features warped *farther* from their keyframe. Thus, when $p$ is small relative to $n$, we weight the previous keyframe's features more heavily, and vice versa. In summary, the features for intermediate frame $I_i$ are set to: $f_i = F(\frac{n-p}{n} f^f, \frac{p}{n} f^b)$, where $p = i \mod n$. This scheme is reflected in Alg. 2.

## 4 EXPERIMENTS

### 4.0.1 DATASETS

We train and evaluate our system on CamVid (Brostow et al. (2009)) and Cityscapes (Cordts et al. (2016)), two popular, large-scale datasets for complex urban scene understanding. CamVid consists

of over 10 minutes of footage captured at 30 fps and $960 \times 720$ pixels. Cityscapes consists of 30-frame video snippets shot at 17 fps and $2048 \times 1024$ pixels. On CamVid, we adopt the standard train-test split of Sturgess et al. (2009). On Cityscapes, we train on the `train` split and evaluate on the `val` split, following the example of previous work (Yu et al. (2017); Chen et al. (2017); Zhu et al. (2017)). We use the standard mean intersection-over-union (mIoU) metric to evaluate segmentation accuracy, and measure throughput in frames per second (fps) to evaluate inference performance.

### 4.0.2 ARCHITECTURE

For our segmentation network, we adopt a variant of the DeepLab architecture called Deformable DeepLab (Dai et al. (2017)), which employs *deformable convolutions* in the last ResNet block (conv5) to achieve significantly higher accuracy at comparable inference cost to a standard DeepLab model. DeepLab (Chen et al. (2017)) is widely considered a state-of-the-art architecture for semantic segmentation, and a DeepLab implementation currently ranks first on the PASCAL VOC object segmentation challenge (Aytar (2018)). Our DeepLab model uses ResNet-101 as its feature network, which produces intermediate representations $f_i \in R^{1 \times 2048 \times \frac{h}{16} \times \frac{w}{16}}$. The DeepLab task network outputs predictions $p_i \in R^{1 \times C \times h \times w}$, where $C$ is 12 or 20 for CamVid and Cityscapes respectively.

### 4.0.3 TRAINING

To train our single-frame DeepLab model, we initialize with an ImageNet-trained ResNet-101 model, and learn task-specific weights on the CamVid and Cityscapes `train` sets. To train our video segmentation system, we sample at random a labeled image from the train set, and select a preceding and succeeding frame to serve as the previous and next keyframe, respectively. Since motion estimation with block motion vectors and feature warping are both parameter-free, feature propagation introduces no additional weights. Training feature interpolation with convolutional fusion, however, involves learning weights for the 1x1 conv fusion layer, which is applied to stacked feature maps, each with channel dimension 2048. For both schemes, we train with SGD on an AWS EC2 instance with 4 Tesla K80 GPUs for 50 epochs, starting with a learning rate of $10^{-3}$.

## 4.1 RESULTS

### 4.1.1 BASELINE

For our accuracy and performance baseline, we evaluate our full DeepLab model on every labeled frame in the CamVid and Cityscapes `test` splits. Our baseline achieves an accuracy of 68.6 mIoU on CamVid, at a throughput of 3.7 fps. On Cityscapes, the baseline model achieves 75.2 mIoU, matching published results for the DeepLab architecture we used (Dai et al. (2017)), at 1.3 fps.

### 4.1.2 PROPAGATION AND INTERPOLATION

In this section, we evaluate our two main contributions: 1) feature propagation with block motion vectors (**prop-BMV**), and 2) feature interpolation, our new feature estimation scheme, implemented with block motion vectors (**inter-BMV**). We compare to the closest available existing work on the problem, a feature propagation scheme based on optical flow (Zhu et al. (2017)) (**prop-flow**). We evaluate by comparing accuracy-runtime curves for the three approaches on CamVid and Cityscapes (see Fig. 3). These curves are generated by plotting accuracy against throughput at each keyframe interval in Appendix: Tables 4 and 5, which contain comprehensive results.

First, we note that block motion-based feature propagation (**prop-BMV**) *outperforms* optical flow-based propagation (prop-flow) at all but the lowest throughputs. While motion vectors are slightly less accurate than optical flow in general, by cutting inference times by 53% on intermediate frames (Sec. 3.3.1), prop-BMV enables operation at much lower keyframe intervals than optical flow to achieve the same inference rates. This results in a much more favorable accuracy-throughput curve.

Second, we find that our feature interpolation scheme (**inter-BMV**) *strictly outperforms* both feature propagation schemes. At every keyframe interval, inter-BMV is more accurate than prop-flow and prop-BMV; moreover, it operates at similar throughput to prop-BMV. This translates to a consistent advantage over prop-BMV, and an even larger advantage over prop-flow (see Fig. 3). On CamVid, inter-BMV actually registers a small *accuracy gain* over the baseline at keyframe intervals 2 and 3, utilizing multi-frame context to improve on the accuracy of the single-frame DeepLab model.

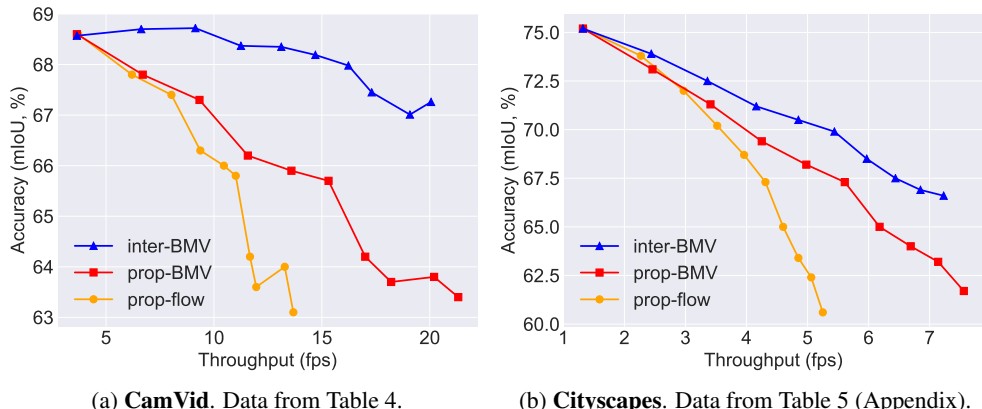

(a) **CamVid**. Data from Table 4.      (b) **Cityscapes**. Data from Table 5 (Appendix).

Figure 3: Accuracy (**avg.**) vs. throughput for all schemes on CamVid and Cityscapes.

**Metrics.** We also distinguish between two metrics: the standard *average* accuracy, results for which are plotted in Fig. 3, and *minimum* accuracy, which is a measure of the lowest frame-level accuracy an approach entails, i.e. accuracy on frames farthest away from keyframes. Minimum accuracy is the appropriate metric to consider when we wish to segment a video as efficiently as possible, while ensuring that all frame segmentations meet some threshold level of accuracy. As an example, at an accuracy target of 65 mIoU, feature interpolation enables operation at 20.1 fps on CamVid (see Table 4). This is $2.5\times$ faster than achievable inference speeds with feature propagation alone, using either optical flow (8.0 fps) or block motion vectors (9.3 fps). In general, feature interpolation achieves over twice the throughput as Zhu et al. (2017) on CamVid and Cityscapes, at any target accuracy. Minimum accuracy plots (Fig. 5) are included in the Appendix.

**Baseline.** We also compare to our frame-by-frame DeepLab baseline, which offers low throughput but high average accuracy. As Figures 3a and 3b indicate, even at average accuracies above 68 mIoU on CamVid and 70 mIoU on Cityscapes, figures competitive with contemporary single-frame models (see Table 1 and Table 2), feature interpolation offers speedups of $4.5\times$ and $4.2\times$, respectively, over the baseline. Notably, at key interval 3, interpolation obtains a $2.5\times$ speedup over the baseline on CamVid, at slightly *higher than baseline accuracy* (see Fig. 3a and Table 4).

Table 1: Comparing **inter-BMV** to various contemporary segmentation networks on CamVid.

| Scheme | Accuracy (mIoU, %) | Throughput (fps) | Model |
|---|---|---|---|
| GRFP (Nilson et al. (2018)) | 66.1 | – | D8+GRFP (best) |
| DFF (Zhu et al. (2017)) | 67.4 | 8.0 | KI=3 |
| LinkNet (Chaurasia et al. (2018)) | 68.3 | – | LinkNet (best) |
| **inter-BMV** | 68.7 | 9.1 | KI=3 |
| DDSC (Bilinski & Prisacariu (2018)) | 70.9 | – | Single scale (best) |

Table 2: Comparing **inter-BMV** to various contemporary segmentation networks on Cityscapes.

| Scheme | Accuracy (mIoU, %) | Throughput (fps) | Model |
|---|---|---|---|
| Clockwork (Shelhamer et al. (2016)) | 64.4 | – | Alternating (best) |
| DRN (Yu et al. (2017)) | 70.9 | – | DRN-C-42 (best) |
| DeepLab-v3 (Chen et al. (2017)) | 71.4 | – | DL-101 (best) |
| DFF (Zhu et al. (2017)) | 72.0 | 3.0 | KI=3 |
| **inter-BMV** | 72.5 | 3.4 | KI=3 |
| RefineNet (Lin et al. (2017)) | 73.6 | – | RN-101 (best) |

**Delay.** Recall that feature interpolation introduces a delay of keyframe interval $n$ frames, which corresponds to $\frac{n}{30}$ seconds at 30 fps. For example, at $n = 3$, inter-BMV introduces a delay of $\frac{3}{30}$

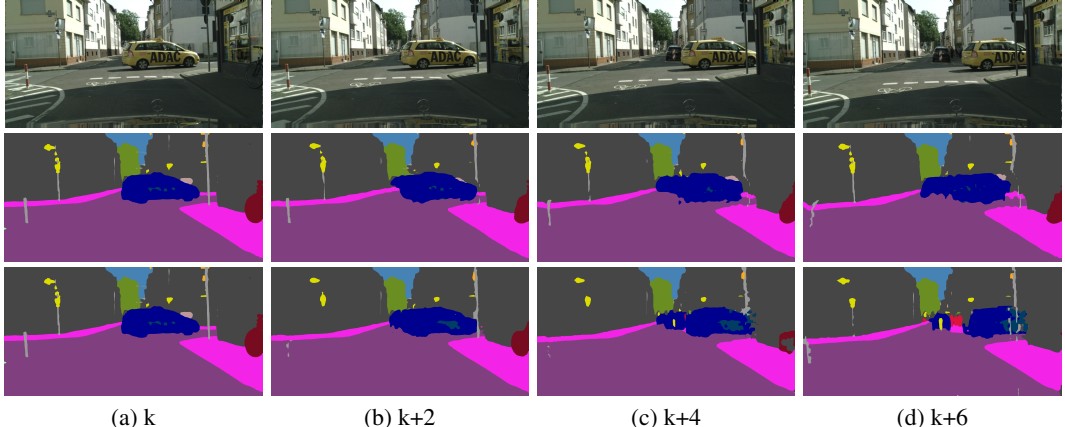

| (a) k | (b) k+2 | (c) k+4 | (d) k+6 |

Figure 4: Example segmentations at keyframe interval 7. Column $k + i$ corresponds to outputs $i$ frames past the selected keyframe $k$. **First row:** input frames. **Second row:** prop-flow (Zhu et al. (2017)). **Third row:** inter-BMV (us). Note that, by $k + 6$, prop-flow has significant warped the moving car, obscuring the people, vehicle, and street sign in the background (image center), while these entities remain clearly visible with interpolation, which exploits full scene context. *Cityscapes.*

seconds, or $100$ ms. To put this in context, prop-flow (Zhu et al. (2017)) takes 125 ms to segment a frame at key interval 3, and inter-BMV takes 110 ms. Thus, by lagging by less than 1 segmentation, we are able to segment $2.5\times$ more frames per hour than the frame-by-frame model (9.1 fps vs. 3.6 fps). This is a suitable tradeoff in almost all batch settings (e.g. training data generation, post-hoc video analysis), and in many interactive applications (e.g. video anomaly detection, film editing).

Fig. 4 depicts a **qualitative comparison** of interpolation and prop-flow (Zhu et al. (2017)).

### 4.1.3 FEATURE FUSION

In this second set of experiments, we evaluate the accuracy gain achieved by feature fusion, in order to isolate the contribution of fusion to the success of our feature interpolation scheme. As Table 3 demonstrates, utilizing any fusion strategy, whether max, average, or conv fusion, results in higher accuracy than using either input feature map alone. This holds true even when one feature map is significantly stronger than the other (rows 2-4), and for both short and long distances to the keyframes. This observed additive effect suggests that feature fusion is highly effective at capturing signal that appears in only one input feature map, and in merging spatial information across time.

Table 3: An evaluation of feature fusion. We report final accuracies for various keyframe placements. Forward and Backward refer to the input feature maps (and their direction of warping). *Cityscapes.*

| Distance to keyframe(s) | Forward mIoU | Backward mIoU | Max Fusion mIoU | Avg. Fusion mIoU | Conv. Fusion mIoU |
|---|---|---|---|---|---|
| 1 | 71.8 | 69.9 | 72.6 | 72.8 | 72.6 |
| 2 | 67.8 | 62.4 | 68.2 | 68.5 | 68.2 |
| 3 | 64.9 | 59.8 | 66.3 | 66.7 | 66.4 |
| 4 | 62.4 | 57.3 | 64.5 | 65.0 | 64.7 |

## 5 CONCLUSION

We develop **interpolation-BMV**, a novel segmentation scheme that combines the use of block motion vectors for feature warping, bi-directional propagation to capture scene context, and feature fusion to produce accurate frame segmentations at high throughput. We evaluate on the CamVid and Cityscapes datasets, and demonstrate significant speedups across a range of accuracy levels, compared to both a strong single-frame baseline and prior work. Our methods are general, and represent an important advance in the effort to operate image models efficiently on video.

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

## 6 APPENDIX

### 6.1 SYSTEM DESIGN

We provide the formal statement of feature interpolation, Algorithm 2.

---

**Algorithm 2** Feature interpolation with block motion vectors (**inter-BMV**)

---
1: **input**: video frames $\{I_i\}$, motion vectors mv, keyframe interval $n$
2: $W_f, W_b \leftarrow []$               ▷ forward, backward warped features
3: **for** frame $I_i$ **in** $\{I_i\}$ **do**
4:    **if** i **mod** n == 0 **then**               ▷ keyframe
5:      $f_i \leftarrow N_{feat}(I_i)$             ▷ curr keyframe features
6:      $S_i \leftarrow N_{task}(f_i)$
7:      $f_{i+n} \leftarrow N_{feat}(I_{i+n})$          ▷ next keyframe features
8:      $W_f \leftarrow \text{PROPAGATE}(f_i, n - 1, -\text{mv}[i + 1 : i + n])$
9:      $W_b \leftarrow \text{PROPAGATE}(f_{i+n}, n - 1, \text{mv}[i + n : i + 1])$
10:    **else**                  ▷ intermediate frame
11:      $p \leftarrow$ i **mod** n           ▷ offset from prev keyframe
12:      $f_i \leftarrow F(\frac{n-p}{n} \cdot W_f[\text{p}], \frac{p}{n} \cdot W_b[n - \text{p}])$    ▷ fuse propagated features
13:      $S_i \leftarrow N_{task}(f_i)$
14:    **end if**
15: **end for**
16: **output**: frame segmentations $\{S_i\}$

17: **function** PROPAGATE(features $f$, steps $n$, warp array $g$)    ▷ warp $f$ for $n$ steps with $g$
18:    $O \leftarrow [f]$
19:    **for** i = 1 **to** n **do**
20:      append($O$, WARP($O[i - 1], g[i])$)        ▷ warp features one step
21:    **end for**
22:    **return** $O$
23: **end function**

---

### 6.2 RESULTS

This appendix section includes full tabular results for the CamVid and Cityscapes datasets (**Table 4** and **Table 5**) and minimum accuracy vs. throughput plots for CamVid and Cityscapes (**Figure 5**).

Table 4: Accuracy and throughput on **CamVid** for three schemes: (1) feature propagation with optical flow (Zhu et al. (2017)) (**prop-flow**), (2) feature propagation with block motion vectors (**prop-BMV**), and (3) feature interpolation with block motion vectors (**inter-BMV**).

| Metric | Scheme | keyframe interval | | | | | | | | | |
| | | 1 | 2 | 3 | 4 | 5 | 6 | 7 | 8 | 9 | 10 |
|---|---|---|---|---|---|---|---|---|---|---|---|
| mIoU, avg | prop-flow | 68.6 | 67.8 | 67.4 | 66.3 | 66.0 | 65.8 | 64.2 | 63.6 | 64.0 | 63.1 |
| (%) | prop-BMV | 68.6 | 67.8 | 67.3 | 66.2 | 65.9 | 65.7 | 64.2 | 63.7 | 63.8 | 63.4 |
| | inter-BMV | **68.6** | **68.7** | **68.7** | **68.4** | **68.4** | **68.2** | **68.0** | **67.5** | **67.0** | **67.3** |
| mIoU, min | prop-flow | 68.5 | 67.0 | 66.2 | 64.9 | 63.6 | 62.7 | 61.3 | 60.5 | 59.7 | 58.7 |
| (%) | prop-BMV | 68.5 | 67.0 | 65.9 | 64.7 | 63.4 | 62.7 | 61.4 | 60.8 | 60.0 | 59.3 |
| | inter-BMV | **68.5** | **68.6** | **68.4** | **68.2** | **67.9** | **67.4** | **67.0** | **66.4** | **66.1** | **65.7** |
| throughput | prop-flow | 3.6 | 6.2 | 8.0 | 9.4 | 10.5 | 11.0 | 11.7 | 12.0 | 13.3 | 13.7 |
| (fps) | prop-BMV | 3.6 | 6.7 | 9.3 | 11.6 | 13.6 | 15.3 | 17.0 | 18.2 | 20.2 | 21.3 |
| | inter-BMV | 3.6 | 6.6 | 9.1 | 11.3 | 13.1 | 14.7 | 16.2 | 17.3 | 19.1 | 20.1 |

Table 5: Accuracy and throughput on **Cityscapes** for the three schemes: prop-flow (Zhu et al. (2017)), prop-BMV, and inter-BMV.

| Metric | Scheme | keyframe interval | | | | | | | | | |
|---|---|---|---|---|---|---|---|---|---|---|---|
| | | 1 | 2 | 3 | 4 | 5 | 6 | 7 | 8 | 9 | 10 |
| mIoU, avg | prop-flow | 75.2 | 73.8 | 72.0 | 70.2 | 68.7 | 67.3 | 65.0 | 63.4 | 62.4 | 60.6 |
| (%) | prop-BMV | 75.2 | 73.1 | 71.3 | 69.4 | 68.2 | 67.3 | 65.0 | 64.0 | 63.2 | 61.7 |
| | inter-BMV | **75.2** | **73.9** | **72.5** | **71.2** | **70.5** | **69.9** | **68.5** | **67.5** | **66.9** | **66.6** |
| mIoU, min | prop-flow | 75.2 | 72.4 | 68.9 | 65.6 | 62.4 | 59.1 | 56.3 | 54.4 | 52.5 | 50.5 |
| (%) | prop-BMV | 75.2 | 71.3 | 67.7 | 64.8 | 62.4 | 60.1 | 58.5 | 56.9 | 55.0 | 53.7 |
| | inter-BMV | **75.2** | **72.5** | **71.5** | **68.0** | **67.2** | **66.2** | **65.4** | **64.6** | **63.5** | **62.9** |
| throughput | prop-flow | 1.3 | 2.3 | 3.0 | 3.5 | 4.0 | 4.3 | 4.6 | 4.9 | 5.1 | 5.3 |
| (fps) | prop-BMV | 1.3 | 2.5 | 3.4 | 4.3 | 5.0 | 5.6 | 6.2 | 6.7 | 7.1 | 7.6 |
| | inter-BMV | 1.3 | 2.4 | 3.4 | 4.2 | 4.9 | 5.4 | 6.0 | 6.4 | 6.9 | 7.2 |

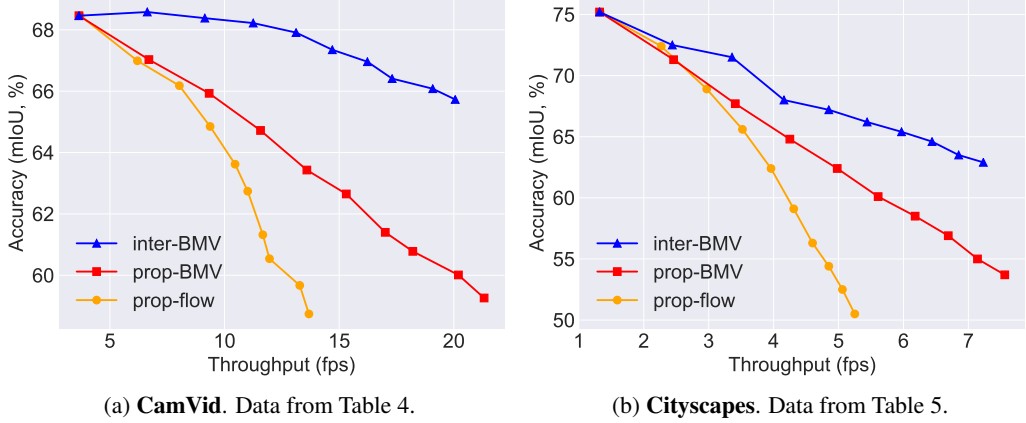

(a) **CamVid**. Data from Table 4.    (b) **Cityscapes**. Data from Table 5.

Figure 5: Accuracy (**min.**) vs. throughput for all schemes on CamVid and Cityscapes.

