# OpenReview forum: "Inter-BMV: Interpolation with Block Motion Vectors for Fast Semantic Segmentation on Video"
_ICLR.cc/2019/Conference_

### Official Review · AnonReviewer1 · 2018-11-02
**Encouraging results but main idea is not novel and some baselines are missing**

**Rating:** 5
**Confidence:** 4

**Review:**

# Paper summary
This paper advances a method for accelerating semantic segmentation on video content at higher resolutions. Semantic segmentation is typically performed over single images, while there is un-used redundancy between neighbouring frames. The authors propose exploiting this redundancy and leverage block motion vectors from MPEG H.264 video codec which encodes residual content between keyframes. The block motion vectors from H264 are here used to propagate feature maps from keyframes to neighbouring non-keyframe frames (in both temporal directions) avoiding thus an additional full forward pass through the network and integrate this in the training pipeline. Experimental results on CamVid and Cityscapes show that the proposed method gets competitive results while saving computational time.


# Paper strengths
- This paper addresses a problem of interest for both academic and industrial purposes.
- The paper is clearly written and the authors argument well their contributions, adding relevant plots and qualitative results where necessary.
- The two-way interpolation with block motion vectors and the fusion of interpolated features are novel and seem effective.
- The experimental results, in particular for the two-way BMV interpolation, are encouraging.


# Paper weaknesses

- The idea of using Block Motion Vectors from compressed videos (x264, xvid) to capture motion with low-cost has been previously proposed and studied by Kantorov and Laptev [i] in the context of human action recognition. Flow vectors are obtained with bilinear interpolation from motion blocks between neighbouring frames. Vectors are then encoded in Fisher vectors and not used with CNNs as done in this paper. In both works, block motion vectors are used as low-cost alternatives to dense optical flow. I would suggest to cite this work and discuss similarities and differences.


- Regarding the evaluation of the method, some recent methods dealing with video semantic segmentation, also using ResNet101 as backbone, are missing, e.g. low latency video semantic segmentation[ii]. Pioneer Clockwork convnets are also a worthy baseline in particular in terms of computational time (results and running times on CityScapes are shown in [ii]). It would be useful to include and compare against them.

- In Section 4.1.2 page 7 the authors mention a few recent single-frame models ((Yu et al. (2017); Chen et al. (2017); Lin et al. (2017); Bilinski & Prisacariu (2018)) as SOTA methods and the current method is competitive with them. However I do not see the results from the mentioned papers in the referenced Figures. Is this intended?

- On a more general note related to this family of approaches, I feel that their evaluation is usually not fully eloquent. Authors compare against similar pipelines for static processing and show gains in terms of computation time. The backbone architecture, ResNet-101 is already costly for high-resolution inputs to begin with and avoiding a full-forward pass brings quite some gains (though a part of this gain is subsequently attenuated by the latency caused by the batch processing of the videos). There are recent works in semantic segmentation that focus on architectures with less FLOPs or memory requirements than ResNet101, e.g. Dilated ResNets [iii], LinkNet[iv]. So it could be expected that image-based pipelines to be getting similar or better performance in less time. I expect the computational gain on such architectures when using the proposed video processing method to be lower than for ResNet101, and it would make the decision of switching to video processing or staying with frame-based predictions more complex.
The advantage of static image processing is simpler processing pipelines at test time without extra parameters to tune. It would be interesting and useful to compare with such approaches on more even grounds.


# Conclusion
This paper takes on an interesting problem and achieves interesting results. The use of Block Motion Vectors has been proposed before in [i] and the main novelty of the paper remains only the interpolation of feature maps using BMVC. The experimental section is missing some recent related methods to benchmark against.
This work has several strong and weak points. I'm currently on the fence regarding my decision. For now I'm rating this work between Weak Reject and Borderline

# References

[i] V. Kantorov and I. Laptev, Efficient feature extraction, aggregation and classification for action recognition, CVPR 2014
[ii] Y. Li et al., Low-Latency Video Semantic Segmentation, CVPR 2018
[iii] F. Yu et al., Dilated Residual Networks, CVPR 2017
[iv] A. Chaurasia and E. Culurciello, LinkNet: Exploiting Encoder Representations for Efficient Semantic Segmentation, arXiv 2017

---

> ### Author Response · Authors · 2018-11-10
> **Author response**
>
> Thanks very much for your thoughtful comments on our paper.
>
> (1) Thanks for pointing our attention to Kantorov and Laptev 2014. While Kantorov and Laptev do explore MPEG block motion vectors, they do so in a very different context, treating motion vectors as low-level video features (“descriptors”) to learn more effectively on video. This is a very similar idea to that proposed in CoViAR [i], which trains directly on video I-frames, motion vectors, and residuals (also in the context of action recognition). CoViAR (Wu et al. 2018) is cited and discussed in our paper (Sec. 2.3):
>
>      “Wu et al. (2018) train a network directly on compressed video to improve both accuracy and performance on video
>      action recognition... Unlike these works, our main focus is not efficient training, nor reducing the physical size of
>      input data to strengthen the underlying signal for video-level tasks, such as action recognition.”
>
> In contrast, we center our efforts on efficient, frame-level inference (Sec. 2.3 cont’d):
>
>      “We instead focus on a class of dense prediction tasks, notably semantic segmentation, that involve high-
>       dimensional output (e.g. a class prediction for every pixel in an image) generated on the original uncompressed
>       frames of a video. This means that we must still process each frame in isolation. To the best of our knowledge, we
>       are the first to propose the use of compressed video artifacts to warp deep neural representations, with the goal of...
>       improved inference throughput on realistic video.”
>
> We will add a citation to Kantorov and Laptev 2014 in our paper revision. Thanks once again for the reference.
>
> [i] Wu et al. Compressed Video Action Recognition. CVPR 2018.
>
>
> (2) We compare to these methods in the table below.
>
>
> (3) Here are comparisons to CC, the single-frame models we cited, and other SoA methods. Note that even while none of these schemes report inference times, we still outperform (or are competitive) on accuracy. We’d be happy to include this table in the revised paper, if helpful.
>
> Cityscapes		        Accuracy (mIoU)	 Throughput (fps)     Model notes
> DFF [i]			                 72.0			    3.0		     KI=3*
> Inter-BMV (us)			 72.5			    3.4		     KI=3
>
> Clockwork (2016) [ii]		 64.4			     --		     Alternating (best)
> Yu et al. (2017)		         70.9			     --		     DRN-C-42 (best)
> Chen et al. (2017)		         71.4			     --	             DL-101 (best)
> Lin et al. (2017)		         73.6			     --		     RN-101 (best)
>
> CamVid  			Accuracy (mIoU)	 Throughput (fps)     Notes
> DFF [i]			                 67.4			    8.0		     KI=3
> Inter-BMV (us)			 68.7			    9.1		     KI=3
>
> GRFP (2018) [iii]		         66.1			     --		     D8+GRFP (best)
> LinkNet (2017) [iv]		 68.3			     --		     LinkNet (best)
> Bilinski et al. (2018)		 70.9			     --	             Single scale (best)
>
> *KI = keyframe interval
>
> [i] Zhu et al. Deep Feature Flow for Video Recognition. CVPR 2017.
> [ii] Shelhamer et al. Clockwork Convnets for Video Semantic Segmentation. ECCV Workshops 2016.
> [iii] D. Nilsson and C. Sminchisescu. Semantic video segmentation by gated recurrent flow propagation. CVPR 2018.
> [iv] A. Chaurasia and E. Culurciello. LinkNet: exploiting encoder representations for efficient semantic segmentation. arXiv 2017.
>
>
> (4) This is a good suggestion. We compare against Dilated ResNets (Yu et al. 2017) and LinkNet (Chaurasia et al. 2017) in the previous table.
>
> Thanks once again for the taking the time to review our paper. We look forward to hearing back!

---

> > ### Comment · AnonReviewer1 · 2018-11-28
> > **Final comments**
> >
> > Dear all,
> >
> > I went through the answers from the authors and the opinions of the other reviewers. The authors provided an elaborated rebuttal with additional clarifications and experiments. The authors position themselves well w.r.t. to MPEG-flow and provide additional baselines.
> >
> > However, my concern regarding the relevance of this work in modern architectures which are lighter and powerful and can get fast predictions from frames only have not been adressed. The authors compare against LinkNet and DRN in terms of accuracy, but not in terms of throughput. Taking the results from the LinkNet paper, for images of size 1920 x 1080 on a NVIDIA Titan X GPU (relatively similar conditions with the current work), they reach 8.5 fps. The current pipeline which is more complex and not as easy to deploy does 9.1 fps. Of course, inter-BMV can do faster while sacrificing accuracy, but as I mentioned in my review, in practice the trade-off and decision of switching to video processing are not obvious.
> >
> > Wrapping up, there are some nice ideas and results in the current paper, but I am not convinced for accepting it to the conference. I think this would be a very good contribution for the workshop track.

---

### Official Review · AnonReviewer3 · 2018-11-05
**The paper presents a feature interpolation strategy that  has limited novelty**

**Rating:** 3
**Confidence:** 5

**Review:**

This paper presents a feature interpolation strategy for fast semantic segmentation in videos. They first compute features of keyframes, then interpolate intermediate frames based on block-motion vectors (BMV), and finally fuse the interpolated features as input to the prediction network. The experiments show that the model outperforms one recent, closely related work wrt inference time while preserving accuracy.

Positive:
1. Efficient inference. The strategy cuts inference time on intermediate frames by 53%, while achieves better accuracy and IOU compared to the one recent closely related work.

2. The ablation study seems sufficient and well-designed. The paper presents two feature propagation strategies and three feature fusion methods. The experiments compare these different settings, and show that interpolation-BMV is indeed a better feature propagation.

Negative:

1. Limited novelty. The algorithm is close to the optical-flow based models Shelhamer et al. (2016) and Zhu et al. (2017). The main difference is that the optical-flow is replaced with BMV, which is a byproduct of modern cameras.

2. Insufficient experimental comparison with other baselines. In experiments, the paper compares the proposed model with only one baseline Prop-flow, which is not a sufficient comparison to show that the paper really outperforms the state-of-art model. For example, the authors should also compare with “Clockwork convnets for video semantic segmentation.”

3. Some technical details are not clear. For example, in section 3.1, the paper mentions that the task network is built by concatenating three components but never clarifies them. Also, in algorithm 2, line 13 shows that F is a function with two entries, but line 8 indicates that F is a feature.

---

> ### Author Response · Authors · 2018-11-10
> **Author response**
>
> Thanks a lot for taking the time to review our paper.
>
> (1) Limited novelty -- Our paper is very keen on the distinction between our work and Shelhamer et al. (2016) and Zhu et al. (2017). First, Shelhamer et al. (2016) does not use optical flow, and instead simply copies features from frame to frame (and schedules this copying). Zhu et al. (2017) then proposes an improvement to this scheme, forward feature warping with optical flow. In general, both these techniques fail to achieve speedups beyond small multiples of the baseline (< 3x), without impacting accuracy. The key reason for this is that both feature copying and forward warping are unable to capture *new scene content*. In fast moving footage (e.g. driving footage), copied and warped features quickly become obsolete, and warping error compounds significantly (see e.g. qualitative outputs, Fig. 8, in our paper).
>
> In Inter-BMV, we exploit the observation that scenes tend to have semantic start and end points -- e.g. a pedestrian walking across a crosswalk, a car turning a street corner. This allows us to leverage bi-directional warping, a new idea, to strong effect. Our second insight -- that video is compressed by default in a temporally referential manner (e.g. P-/B-frames in H.264 video) -- lends itself to an alternate, computation-free motion estimation scheme. This, together with our observation that video can be more efficiently processed in mini-batches, e.g. of 10 frames, enables us to trade-off a small amount of latency for a large gain in throughput. Ten video frames consists of 330 ms of footage at 30 fps -- this is comparable to the human visual reaction time (230-400 ms, see studies [i]), yet allows us to accelerate segmentation by almost *6x* over frame-by-frame, while maintaining within 1-2% of baseline accuracy.
>
> To the best of our knowledge, none of our core ideas -- (1) bi-directional feature warping, (2) the use of block motion vectors for deep representation warping, and (3) mini-batch processing of video to accelerate segmentation throughput -- have been proposed or published before.
>
> [i] https://www.ncbi.nlm.nih.gov/pmc/articles/PMC4374455/
>
>
> (2) Here are comparisons with other SoAs (ranked by accuracy). Note that we significantly outperform Clockwork Convnets (Shelhamer et al. 2016). Note also that CC does not report results on CamVid, nor does it report inference times.
>
> Cityscapes			Accuracy (mIoU)	Throughput (fps)	  Key interval
> Clockwork [i]			          64.4			 --			           2
> DFF [ii]				          68.7			 4.0			           5
> GRFP [iii]			          69.4			 2.1			           5
> Inter-BMV (us)			  70.5			 4.9			           5
>
> CamVid			        Accuracy (mIoU)	Throughput (fps)	  Key interval
> GRFP [iii]			          66.1			 --		                   --
> DFF [i]				          67.4			 8.0			           3
> Inter-BMV (us)			  68.7			 9.1			           3
>
> [i] Shelhamer et al. Clockwork Convnets for Video Semantic Segmentation. ECCV Workshops 2016.
> [ii] Zhu et al. Deep Feature Flow for Video Recognition. CVPR 2017.
> [iii] D. Nilsson and C. Sminchisescu. Semantic video segmentation by gated recurrent flow propagation. CVPR 2018.
>
> For a more extensive comparison with a number of other segmentation architectures, please see our response to AnonReviewer1!
>
>
> (3) We describe our task network as follows (Sec. 3.1, p. 3):
>
>      “We identify two logical components in our final model: a feature network, which takes as input an image i ∈
>      R^{1×3×h×w} and outputs a representation f_i ∈ R^{1×A×h/16×w/16}, and a task network, which given the
>      representation, computes class predictions for each pixel in the image, p_i ∈ R^{1×C×h×w}.
>
>      The task network N_task is built by concatenating three blocks: (1) a feature projection block, which reduces the
>      feature channel dimensionality to A/2, (2) a scoring block, which predicts scores for each of the C segmentation
>      classes, and (3) an upsampling block, which bilinearly upsamples the score maps to the resolution of the input
>      image.”
>
> We used the DeepLab segmentation architecture (Chen et al. 2017), so we omitted further details about the task network, provided here:
>      (1) Feature projection block - R^{1×A×h/16×w/16} -> R^{1×A/2×h/16×w/16}
>      (2) Scoring block - R^{1×A/2×h/16×w/16} -> R^{1×C×h/16×w/16}
>      (3) Upsampling block - R^{1×C×h/16×w/16} -> R^{1×C×h×w}
>
> Regarding Algorithm 2 in the Appendix, good catch!
> 	Line 8 should read f_{k+n} ← N_{feat} (I_{k+n}) NOT
> 			                  f_{k+n} ← N_{feat} (F_{k+n})
> 	where I_{k+n} refers to the k+n-th frame in the video.
>
> We will correct this in our revision.
>
>
> Thanks a lot once again for your comments. We look forward to your response!

---

### Official Review · AnonReviewer2 · 2018-11-06
**Interesting idea but need further clarified**

**Rating:** 5
**Confidence:** 4

**Review:**

In this paper, the authors propose a novel segmentation scheme that combines the block motion vectors for feature warping, bi-directional propagation, and feature fusion. Experiments demonstrate its effectiveness compared with alternative methods. However, I still have several concern:
1. As  the block motion vectors are generally rough estimation, it may damage the performance of the tasks. The authors should further clarify how the imperfect estimation influence the performance, e.g., the Blocking artifacts.
2. The features are actually abstract representation of an image while the motion vectors are actually obtained via the pixel comparison. The authors should further justify the motion estimation could be used to the latent feature directly.
3.  The authors are expected to conduct more comprehensive experiments. Motion vectors are consistent in the current dataset. The authors are expected to demonstrate when the motion are chaotic.

---

> ### Author Response · Authors · 2018-11-10
> **Author response**
>
> Thanks very much for taking the time to review our paper.
>
> (1) The fact that block motion vectors (BMVs) are rougher motion estimates than optical flow is actually discussed in our results section (Sec. 4.1.2):
>
>      “While motion vectors are slightly less accurate than optical flow in general, by cutting inference times by 53% on
>      intermediate frames (Sec. 3.3.1), prop-BMV enables operation at much lower keyframe intervals than optical flow to
>      achieve the same inference speeds. This results in a much more favorable accuracy-throughput curve.”
>
> As a specific example (from Table 1), to achieve throughput of ~13.5 fps on CamVid requires operating at keyframe interval 10 with prop-flow (63.1 mIoU) but only keyframe interval 5 with prop-BMV (65.9 mIoU), which enables ~3% higher mIoU.
>
> In essence, because motion estimation with block motion vectors is *much* cheaper than motion estimation with optical flow, block motion vectors allow us to operate at lower keyframe intervals, and thus achieve *higher accuracy*, for a given inference speed, than optical flow. This holds even given the small head-to-head accuracy difference between flow and BMV.
>
> This is one of the key findings of our paper, and we would be happy to clarify further.
>
> As for blocking artifacts, these are minor, but visible in our qualitative outputs (Fig. 8) -- for example, optical flow is better at preserving thin details, such as the street sign on the left (yellow in the segmentation output). In contrast, forward flow warping causes drastic distortion of moving objects (e.g. the “ADAC” taxi), occluding objects in the background (e.g. the pedestrians). This is reflected in much lower *overall* quantitative accuracy (mIoU) for prop-flow than for inter-BMV. We will add a note about blocking artifacts to the caption of Fig. 8 in our revision.
>
>
> (2) Our choice is inspired by the use of optical flow, in previous work (e.g. DFF), to warp deep features. Like block motion, optical flow is also computed directly on image pixels (albeit with more complex methods, e.g. Lukas-Kanade [i] or Farneback [ii]), but is still able to effectively warp the intermediate representations of ResNet-based image/video recognition networks. The core reason that pixel-level motion estimates suffice for feature warping is that fully convolutional architectures, such as e.g. FCN [iii] or DeepLab [iv] for segmentation, *preserve spatial structure* in their intermediate representations.
>
> [i] B. Lucas and T. Kanade. An iterative image registration technique with an application to stereo vision. In DARPA Image Understanding Workshop, pages 121–130, 1981.
> [ii] G. Farneback. Two-frame motion estimation based on polynomial expansion. In SCIA, 2003.
> [iii] J. Long et al. Fully convolutional networks for semantic segmentation. CVPR 2015.
> [iv] Chen et al. Rethinking atrous convolution for semantic image segmentation. TPAMI 2017.
>
>
> (3) We evaluated on two datasets (CamVid, Cityscapes). These are the two most popular benchmarks for segmentation research, and are representative of *realistic video*, which demonstrates strong temporal structure (i.e. lack of random motion from frame-to-frame). Our key point is that we exploit this temporal continuity to accelerate segmentation for practical applications, such as video analytics, interactive film editing, and autonomous perception.
>
> To directly address the reviewer’s point that “the authors are expected to demonstrate when the motion [is] chaotic”, our techniques are not dependent on any particular structure in the motion vectors. We apply a well-known warping operator that spatially transforms the features with a bilinear upsampling of the motion vector maps [i]. This operation applies even if the vector maps are highly dense or irregular.
>
> Please also see our responses to other reviewers, which contain e.g. more extensive comparison with other state-of-the-art techniques!
>
> [i] Jaderberg et al. Spatial Transformer Networks. NIPS 2015.
>
>
> Thanks once again for reading through our paper. We look forward to hearing back!

---

### Meta-Review · Area_Chair1 · 2018-12-13
**Some good ideas, but just not rated strongly enough by reviewers**

**Confidence:** 4
**Recommendation:** Reject

**Metareview:**

Strengths:
Paper uses an efficient inference procedure cutting inference time on intermediate frames by 53%, & yields better accuracy and IOU compared to the one recent closely related work.

The ablation study seems sufficient and well-designed. The paper presents two feature propagation strategies and three feature fusion methods. The experiments compare these different settings, and show that interpolation-BMV is indeed a better feature propagation.

Weaknesses: Reviewers believed the work to be of limited novelty. The algorithm is close to the optical-flow based models Shelhamer et al. (2016) and Zhu et al. (2017). Reviewer asserts that the main difference is that the optical-flow is replaced with BMV, which is a byproduct of modern cameras.  R3 felt that there was Insufficient experimental comparison with other baselines and that technical details were not clear enough.

Contention: Authors assert that Shelhamer et al. (2016) does not use optical flow, and instead simply copies features from frame to frame (and schedules this copying). Zhu et al. (2017) then proposes an improvement to this scheme, forward feature warping with optical flow. In general, both these techniques fail to achieve speedups beyond small multiples of the baseline (< 3x), without impacting accuracy.

Consensus: It was disappointing that some of the reviewers did not engage after the author review (perhaps initial impressions were just too low). However, after the author rebuttal R1 did respond and held to the position that the work should not be accepted, justified by the assertion that other modern architectures that are lighter weight and are able to produce fast predictions.